# Recent Advances in Silver Nanoparticles Containing Nanofibers for Chronic Wound Management

**DOI:** 10.3390/polym14193994

**Published:** 2022-09-23

**Authors:** Govindaraj Sabarees, Vadivel Velmurugan, Ganesan Padmini Tamilarasi, Veerachamy Alagarsamy, Viswas Raja Solomon

**Affiliations:** 1Department of Pharmaceutical Chemistry, SRM College of Pharmacy, SRM Institute of Science and Technology, Kattankulathur, Chennai 603203, Tamil Nadu, India; 2Department of Pharmaceutical Analysis, SRM College of Pharmacy, SRM Institute of Science and Technology, Kattankulathur, Chennai 603203, Tamil Nadu, India; 3Medicinal Chemistry Research Laboratory, MNR College of Pharmacy, Gr. Hyderabad, Sangareddy 502294, Telangana, India

**Keywords:** composite nanofibers, silver nanoparticles, wound dressing, drug delivery, wound healing, chronic wound management, skin tissue engineering

## Abstract

Infections are the primary cause of death from burns and diabetic wounds. The clinical difficulty of treating wound infections with conventional antibiotics has progressively increased and reached a critical level, necessitating a paradigm change for enhanced chronic wound care. The most prevalent bacterium linked with these infections is *Staphylococcus aureus*, and the advent of community-associated methicillin-resistant *Staphylococcus aureus* has posed a substantial therapeutic challenge. Most existing wound dressings are ineffective and suffer from constraints such as insufficient antibacterial activity, toxicity, failure to supply enough moisture to the wound, and poor mechanical performance. Using ineffective wound dressings might prolong the healing process of a wound. To meet this requirement, nanoscale scaffolds with their desirable qualities, which include the potential to distribute bioactive agents, a large surface area, enhanced mechanical capabilities, the ability to imitate the extracellular matrix (ECM), and high porosity, have attracted considerable interest. The incorporation of nanoparticles into nanofiber scaffolds constitutes a novel approach to “nanoparticle dressing” that has acquired significant popularity for wound healing. Due to their remarkable antibacterial capabilities, silver nanoparticles are attractive materials for wound healing. This review focuses on the therapeutic applications of nanofiber wound dressings containing Ag-NPs and their potential to revolutionize wound healing.

## 1. Introduction

Wound healing must be seen as a sequence of regulated and interconnected steps, as shown in (Figure 1), such as coagulation, inflammation, deposition, fibroplasia, extracellular matrix differentiation, contraction, remodelling, and epithelialization [1]. Acute wounds should recover in two to three weeks at most, followed by the remodelling stage, which may take up to two years. Scab development and little to no infection are signs of routine healing. When an infection exists, the immune system often fights it off. However, in cases of severe microbial disease, the immune system cannot eradicate the condition, and the wound develops to the chronic stage and fails to heal sequentially in the expected time frame. Chronic wounds are difficult to keep clean and are more vulnerable to bacterial infections [2,3,4,5]. A high level of bacterial invasion distinguishes these wounds, as well as increased inflammation, reduced oxygenation on the subepithelial tissues, damaged fibroblast, and postponed re-epithelialization [6,7]. Leg ulcers, pressure ulcers, diabetic foot ulcers (DFUs), fungating wounds, and other chronic wounds are among the most typical [8,9].

In addition to being a significant financial burden on healthcare systems, wounds provide a social barrier for patients and their families. According to a retrospective review of Medicare enrollees, 8.2 million people in the US were impacted by wounds in 2014. In the wound category, the total cost of care ranged from USD 28.1 billion to USD 96.8 billion. The estimated USD 9–13 billion administrative expenses for DFUs were among the highest. Given that there are already 463 million diabetics worldwide, which is predicted to reach 700 million by 2045, these expenditures are projected to grow [10,11]. If diabetes is not effectively treated, a diabetic person’s chance of having a DFU ranges from 15% to 25%. Due to peripheral neuropathy and uncontrollable foot infections, DFUs have a 14% to 24% likelihood of resulting in lower limb amputations (LLAs). Since amputation patients often have difficulty adjusting to social and demographic situations, LLAs are linked to a worse quality of life.

Additionally, LLAs are linked to a greater death rate in diabetes individuals; recent research found that the 30-day postamputation mortality rate was 9.8%. The patient’s chance of needing more amputations increases if they live beyond the 30-day mark. According to reports, the mortality rates for small LLAs and large LLAs at one and four years after amputation were 18% and 45%, respectively, and 33% and 65%. In addition to other risk factors, including age, renal disease, and other consequences of diabetes, depression has recently been recognized as one of the risk variables leading to increased death rates following significant LLA. Poor antibacterial activity, high toxicity, a lack of ability to deliver adequate moisture to the wound, and a lack of mechanical performance are some of the drawbacks of most of the wound dressings on the market [12].

The inefficient use of wound dressings can slow down wound healing. To address this need, nanoscale scaffolds with desired properties such as the capacity to distribute bioactive compounds, a large surface area, increased mechanical capabilities, ability to mimic the extracellular matrix (ECM), and high porosity have piqued the attention of researchers. At the moment, the inclusion of nanoparticles into nanofiber scaffolds is a revolutionary method of “nanoparticle dressing,” which has gained great appeal for wound healing [13,14,15]. Furthermore, AgNPs enhance wound healing due to the following benefits: The role of AgNPs’ anti-inflammatory properties in accelerating wound healing AgNPs can be easily incorporated into nanofibers and dressings. This is due to elemental silver’s meager bacterial resistance, antibacterial activity toward many bacterial strains, and facile surface modification that aids drug delivery. The simple synthetic process can synthesize it to create diverse shapes and sizes from 2 to 500 nm, which can be synthesized by changing reaction parameters and simple and inexpensive large-scale production methods. This review focuses on the therapeutic applications of nanofiber wound dressings containing AgNPs and their potential to revolutionize wound healing.

## 2. Structure of the Skin

The skin is a semipermeable membrane that serves as a defence against harm and protects the structural integrity of the human body. It is the most crucial multifunctional organ in the human body, performing critical tasks while defending the body’s internal tissues against pathogens and excessive water loss [16,17]. Figure 2 depicts a generic skin schematic. The epidermis, dermis, and hypodermis are the three primary layers of the skin. The significant functions of the epidermis are to prevent pathogenic microorganisms from entering the body and to keep the body hydrated. The stratum comprises five layers: basale, spinosum, granulosum, lucidum, and corneum, forming the skin epidermal layer. These layers include a variety of non-immune and immune cells (such as Merkel cells, melanocytes, keratinocytes, Langerhans cells, and stem cells). Keratinocytes, which contain spinous, granular, and outermost stratum corneum layers, are produced by the proliferative component of the epidermis. Some regions of the body, such as the foot soles and skin palms, contain an extra stratum lucidum. The stem cells in the basal epidermis regulate the regeneration of injured skin and the restoration of keratinocytes lost by exfoliation. The dermis layer is placed between the epidermis and hypodermis and comprises blood vessels, nerve cells, collagen, hair roots, mesenchymal stem cells, and lymphatic vessels. The dermis’ primary job is to provide structural hardness to the skin. The hypodermis is the subcutaneous fat layer that supports the epidermal and dermal layers and comprises fibroblasts, adipocytes, vasculature, macrophages, and nerves [18,19].

## 3. Brief Medical History of Silver Nanoparticles

Silver nanoparticles have been extensively used in various medicinal applications, as shown in (Figure 3). Silver nanoparticles are already widely used in wound dressings and burn treatment in biomedicine and also in the food and textile industries, in paints, household items, catheters, implants, and cosmetics, as well as in combination with a variety of materials to prevent infection [20,21,22,23,24,25]. During World War I, they treated soldiers’ wounds with silver leaves to stop infections and help them heal [26,27]. Silver as an ion was used in earlier civilizations, particularly in Egypt. This silver ion was used mainly in wound dressings to treat wounds that were hard to heal. In 1998, Ziehl-Abegg was the first firm to introduce AgNPs into wound dressings, resulting in the AgNPs antimicrobial dressing ActicoatTM. Silver products improve efficacy compared to standard wound dressing [28,29]. Acticoat and Actisorb, two silver-based wound dressings, are commercially available [30]. Dermatology increasingly employs metal nanoparticles to expedite wound healing and to treat and prevent bacterial infections [31].

Indeed, AgNPs’ excellent antimicrobial properties have already been tested against 650 bacterial strains [32,33,34]. Because of their extensive antimicrobial properties and ability to reduce the chance of infection from antibiotic-resistant strains, silver nanoparticles also found their application in wound management products [35,36,37,38,39]. AgNPs are utilized in wound care to prevent secondary infections since they are effective against a spectrum of microbes that can slow down the healing process [40]. Their size and charge allow them to enter microorganisms [41,42,43]. The AgNPs destroy many multi-resistant strains. It also helps rid pathogenic microbes that can slow or stop the typical stages of wound healing [44,45].

The biosynthesis of AgNP utilizing aqueous *Bryonia laciniosa* leaf extract resulted in rapid wound epithelialization and scarless wound repair without significant inflammation due to effective cytokine modulation. It was used as a wound healing agent due to its outstanding anti-inflammatory and antibacterial properties [46]. Biosynthesized AgNPs are made from *Delonix elata* leaf aqueous extract for wound healing in human patients who have had anorectal surgery [47]. Due to their sustained ability to release silver ions, which exhibit concentration-dependent toxicity in HaCaT cells, these nanoparticles have a high potential for usage in dermatology and wound healing [48]. Because AgNPs are biocompatible and can avoid this involuntary inflammatory action, they are not nullified by the immune system; thus, they could be used as anti-inflammatory agents [49]. Because of their anti-inflammatory properties, topical application of AgNPs at the wound site minimizes the release of inflammatory cytokines, lymphocytes, and mast cell infiltration, promoting wound healing with minimal scarring. A. Hebeish et al. also evaluated the anti-inflammatory effects of Ag by comparing them to indomethacin. This commercially available anti-inflammatory drug revealed a dose-dependent reduction in inflammation inside the rat bow edema model [50].

A study showed that AgNPs eradicate 44 strains of six fungi [51]. Gajbhiye et al. discovered that biogenic AgNPs were effective against *Fusarium semitectum*, *Pleospora herbarum*, *Trichoderma* spp, *Phoma glomerata*, and *Candida albicans*. They also reported a synergic effect of AgNPs in conjunction with fluconazole [52,53]. When AgNPs were added, they changed the growth rates of all tested fungi except Mortierella spp, meaning that Chaetomium and Stachybotrys could not grow on gypsum products.

J.L. Speshock et al. investigated AgNPs’ potential in prokaryotic and eukaryotic organisms, and AgNPs at about 25 nm or even less were found to have exceptional potential for viral infection suppression [54]. AgNPs inhibited virus attachment, cell penetration, and the cell’s ability to propagate the virus [55]. As an HIV-1 antiviral nanohybrid and in the deactivation of SARS-Cov-2 spike proteins, TPU-Ag worked better than PVA-Ag. TPU-Ag and PVA-Ag nanofibrous membranes displayed increased antibacterial activity by increasing Ag content from 2 to 4 wt. Additionally, the developed membranes showed good mechanical and physical properties and antiviral and antibacterial activities [56].

Silver synthesized from M. Domestica extracts significantly affected breast cancer MCF-7 cells. In contrast, silver synthesized from *O. Vulgare* aqueous extracts had a dose-dependent effect on the A549 cell line [57,58]. *Moringa olifera* stem bark extract was used to produce AgNPs. K. Vasanth et al. investigated the anticancer properties of these biosynthesized AgNPs. The flow cytometry analysis indicated that ROS generation caused apoptosis in HeLa cells [59]. According to the findings, AgNPs effectively prevent the development of HepG2 cells by inducing apoptosis [60]. Venkatesan et al. found that the human breast cancer cells MDA-MB-231 were killed by chitosan-alginate-biosynthesized AgNPs that were highly permeable (IC_50_ = 4.6 mg) [61]. A recent study found that packed quinazolinone polypyrrole with chitosan silver chloride nanocomposite was active against Ehrlich ascites carcinoma cells [62]. I.M. El-Sherbiny et al. found that Chitosan-silver hybrid nanoparticles cause HepG2 cells to die by turning down the BCL2 gene and the P53 gene [63]. A significant decrease in cyclobutene-pyrimidine-dimer creation demonstrated their chemo-preventive efficacy in HaCaT cells after UVB-induced DNA damage, which has a good potential for avoiding skin cancer [64]. The UVB-protective effectiveness of AgNPs in human keratinocytes is proportional to their size [65]. As a result, pre-treating HaCaT cells with smaller AgNPs (10–40 nm) helped shield skin cells from UVB-induced DNA damage and UV-induced apoptosis. Using 60 and 100 nm AgNPs, no prevention was obtained. AgNPs are increasingly used in healthcare and consumer products, so many commercial products now include these nanoparticles for topical administration to human skin.

Over one million people die from malaria yearly, caused by protozoal vector-borne diseases, the most prevalent and dangerous infections in wealthy nations [66]. Z. Jiang et al. are creating novel antimalarial strategies to control the malaria vector. AgNPs were tested against *Plasmodium falciparum* malarial parasites and other antimalarial medications [67]. The bio-reduction of AgNPs was 5%. The malaria vector Anopheles stephensi and chloroquine-sensitive and resistant *P. falciparum* strains were all treated with *Cassia occidentalis* leaf broth [68].

## 4. Wound Healing Properties of Silver Nanoparticles

Compounds of silver, such as silver nitrate and silver sulfadiazine, are often used to treat infections in chronic wounds and burns [69]. AgNPs help fibroblasts change into myofibroblasts, which makes wounds tighter and speeds up the healing of diabetic wounds. AgNPs accelerate wound healing by enhancing keratinocyte proliferation and migration [70,71]. AgNPs may engage with sulphur-containing proteins in bacterial membrane cells and, ideally, attack the respiratory chain, resulting in apoptosis [72]. When it comes into contact with the injured region, they cause neutrophil apoptosis by lowering mitochondrial function, which reduces cytokine production. As a result, the inflammatory response is modulated or reduced, resulting in faster healing [73,74]. However, because of their small size, AgNPs can easily penetrate biofilms and cell membranes, causing DNA damage, inhibiting cell proliferation, and inhibiting cellular ATP production [75]. The silver nanoparticles change the amount of m-RNA in the wound environment. Aside from antibacterial activities, silver surgical textiles exhibit an increase in healing properties; as an outcome, silver exploitation has an optimistic effect on cell migration and proliferation quality [76,77,78]. Cytokine modulation is mediated by silver nanoparticles’ anti-inflammatory activity [79]. As stated in the preceding section, Cytokines can stimulate fibroblasts and chondrocytes to generate ROS [80]. Thus, silver nanoparticle modulation of cytokine production can reduce ROS levels to avoid severe cellular damage and lag wound healing [81]. Silver has many antibacterial effects, making it less likely that bacteria will become resistant and more effective against microorganisms resistant to multiple drugs. When the amount of AgNPs in the dressing increases, the wound area becomes smaller and more collagen is deposited, which is linked to macrophage and fibroblast migration [82,83]. Sustained release mechanisms can decrease silver ion toxicity and stimulate local antibacterial activity [84,85].

## 5. Mechanistic Understanding of Silver Nanoparticles (AgNPs)

The usual quantity of silver in human plasma is less than 2 µg/mL, and this concentration comes from diet and particulate matter inhalation. Oral exposure to silver can also come via dietary supplements, contaminated water, or from eating fish and other aquatic species [86]. Ionic silver can be ingested orally, inhaled, or absorbed through wounds to enter the body. AgNPs are believed to be transported inside the body by two processes: pinocytosis and endocytosis. The development of a revolutionary medication delivery method was prompted by the discovery that nanoscale particles penetrate far deeper than bulk particles. Although the precise mode of action of AgNPs is not yet known, numerous ideas for their antibacterial qualities have been put forth. Its antibacterial effect is thought to solely be caused by the ongoing release of silver in its ionic state [87,88,89]. Silver ions cling to the cytoplasmic membrane and cell wall because of the sulphur protein affinity and electrostatic attraction. This increases the permeability of the membrane and causes the bacterial cell to rupture and degenerate. Reactive oxygen species are produced, and the respiratory enzymes are essentially deactivated when the silver ion enters the bacterial cell. Reactive oxygen species, a critical element in the mechanism of action for silver, contribute significantly to the disruption of the cell membrane and DNA damage (by interacting with sulphur and phosphorus in the DNA molecule), which hinder replication and reproduction and ultimately lead to microbe death. By denaturing ribosomes, silver ions also prevent the synthesis of ATP and hinder the formation of proteins. After anchoring and observing the cell’s surface, silver nanoparticles build up in the cellular wall pits of microorganisms, causing the denaturation and degeneration of the cell membrane. Due to their micro size, they can easily enter cells, rupturing cell organelles and even causing cell lysis. They interfere with the phosphorylation of protein substrates, which can cause cell death and proliferation, which has an impact on the bacterial transduction process as well. Due to their cellular walls being shorter than those of Gram-positive bacteria, Gram-negative bacterial strains are more susceptible to the effects of AgNPs [86,89]. Silver nanoparticles have a significant downside in that bacterial biofilms make them less effective and penetrating. Due to their intricate structure, biofilms typically change the transport chain to shield the membrane from both silver ions and nanoparticles. The nanoparticle size, which is around 50 nm, severely obstructs the path of penetration that is currently being used. Additionally, it has been observed that silver nanoparticle adsorption and deposition on bacterial biofilms reduces the nanoparticles’ ability to diffuse into bacterial cells (Figure 4). Silver’s interaction with a molecule containing a thiol group in bacterial, fungal, and fungus cells provides the basis for silver nanoparticles’ antibacterial effect (Figure 4). It has been observed that bacterial and fungal cells undergo structural changes after coming into touch with silver nanoparticles, albeit the precise process is yet unclear. Silver nanoparticles have higher antibacterial and antifungal characteristics than normal silver particles because of their extensive surface area, which enables better interaction with bacterial and fungal pathogens. Additionally, the gel made of silver nanoparticles penetrates bacteria and fungi in addition to attaching to cell membranes. Silver penetrates cells and connects to the cell membrane and wall, which prevents the cell from respiring [90,91]. When silver is present, Escherichia coli is prevented from absorbing phosphate and from releasing mannitol, succinate, proline, and glutamine. As a result, silver nanoparticles can be used as effective growth inhibitors in a wide range of microbes and are helpful in many antibacterial control systems [92,93].

## 6. Silver Nanoparticles and Their Synthesis

AgNP is synthesized using either a top-down or bottom-up method, as is typical for most nanomaterials, as illustrated in (Figure 5) [94]. Physically breaking down a large complex into smaller parts is a critical component of the top-down strategy. Physical and mechanical procedures such as UV irradiation, lithography, laser ablation, ultrasonic sounds, and photochemical reduction create a strong energy/force to compress the macromolecule into nanoparticles [95]. The physical reduction procedures generally have quick processing durations, yielding AgNPs with a restricted size distribution range. However, they have several drawbacks, including the tremendous amount of equipment space needed, the extended time required to obtain thermal stability, the energy-intensive nature of the methods, and the difficulty in easily using them for scaling-up reasons [96]. When creating nanoparticles from the bottom up, tiny particles such as atoms and molecules are used as the building blocks. These smaller particles come together to form a complex nanoparticle by self-assembly or aided assembly. Chemical, microwave, and biological approaches are subcategories of the bottom-up strategy. For the manufacture of AgNPs in solution, chemical techniques are often utilized. They are produced in water or other organic solvents [97]. Chemical processes are quick and practical but represent a severe environmental risk because they use hazardous substances as reducing agents and create dangerous byproducts.

New alternative techniques that are cost-efficient, energy-efficient, and environmentally benign are swiftly becoming available to counteract these negative environmental consequences. There has been a thorough literature assessment on the development of green or biological synthesis [98,99,100,101]. In green synthesis, an ecologically friendly substance such as microbial (fungal and bacterial) enzymes and phytochemicals from plant extracts are used instead of the capping/stabilizing and reducing agents used in chemical reduction procedures (leaves, roots, barks, flowers, fruits, peels, and seeds). These biological processes generate biocompatible nanoparticles suitable for pharmaceutical and biomedical applications [102,103]. While utilizing microbes as reducing agents in nanoparticle manufacturing is quite tricky and involves several methods, it is more advantageous than chemical and physical techniques. Because they are widely accessible, simple to extract, and do not need laborious processes, plant extracts are considered a solution to the issues mentioned above in the manufacturing of AgNP. For the most part, because they are so readily available, plant extracts are employed to make nanoparticles. Manufacturing AgNPs with regular shapes and sizes is the main difficulty with plant-based synthesis, and the reduction processes of biological approaches are not well understood [104,105,106,107,108].

## 7. Electrospinning

An electrospinning machine consists of four main parts: a high-power source, a syringe pump, a syringe needle carrying solutions, and a fiber deposition collector, as shown in (Figure 6). A positive electrode is attached to the needle, and a negative electrode is connected to the collector to create the applied electric field. As a result, the repulsion charge accumulates near the hemispherical needle tip when a voltage is applied. A Taylor cone is made when the repulsive charge surpasses the surface tension. The negative electrode, which serves as the collector in this procedure, is where the polymer solution is directed to create fibers. The polymer solution evaporates, leaving behind dry fibers ranging in size from nanometers to micrometers deposited on the collector [109,110].

As shown in (Figure 7 and Figure 8), electrospun nanofiber mats are an excellent choice for chronic wound healing because of their numerous advantages and inherent properties. In multiple medical applications, nanofiber composite materials are widely used. Silver nanoparticle-loaded electrospun nanofiber scaffolds showed exceptional antibacterial activity, high porosity, non-toxicity, and biodegradability. Due to their hydrophilic qualities and prolonged release pattern, these nanofiber scaffolds have become more and more well-liked on a global scale. With the help of this delivery method, a better dressing and therapy for ulcers and wounds in diabetic patients would be made available clinically. Silver nanoparticles have been extensively used in various medicinal applications. Encapsulating or coating nanofibrous scaffolds with metal-based nanoparticles can boost their therapeutic efficacy in wound healing applications [111,112]. Synthetic polymers, particularly those with biodegradable and biocompatible properties, may provide excellent treatment options for severe wounds and burn injuries.

## 8. Cytoprotective Effect of Silver Nanoparticle-Loaded Nanofibers

AgNPs have the potential to spread throughout the body, build up in many organs, and cause significant damage. AgNP bystander effects are being reduced by the development of new fabrication techniquesas shown in (Table 1). To minimize environmental impact, disposal options must be investigated. High nanoparticle loading capacity and controllable release mechanisms are two significant advantages that silver nanoparticle-loaded nanofibers are putting forth. Due to the increased surface area and short diffusion length of electrospun nanofibers, a more excellent Ag+ release is accomplished compared to traditional wound dressing materials.

Additionally, regulated release mechanisms can improve local antibacterial effectiveness while decreasing the dose-related systemic toxicity of silver ions. Silver ions can be released from polymeric nanofibers in a variety of ways, including diffusion caused by the polymer matrix swelling (a swelling-controlled system), polymer breakdown, or a combination of the two. Long-lasting antibacterial activity can be attributed in part to controlled release performance. It will be more effective to incorporate AgNPs into the wound dressings rather than placing them directly on the wound bed. This layer also worked as a sieve to prevent cytotoxicity brought on by the extensive release of AgNPs in the area close to the wound [113,114].

**Table 1 polymers-14-03994-t001:** Various forms of fabricated biomaterials for wound healing applications.

S. No.	Wound Dressing Materials	Fabrication Techniques and Outcomes	Ref.
1	Polyurethane/keratin/AgNP biocomposite mats	**Electrospinning method** The material’s keratin content increased fibroblast cell proliferation while also having strong antibacterial properties. A histological analysis showed that the created biocomposite mat could promote wound healing.	[115]
2	Hyaluronan and PVA embedded-AgNP Hydrogel	**Freeze-thawing method** The hydrogel’s semi-interpenetrating network aided in the AgNPs’ uniform dispersion. The hydrogel may be used as a wound dressing since it had strong antibacterial activity, was biocompatible, had a low swelling index, and was nontoxic.	[116]
3	Genipin-crosslinked chitosan/poly(ethylene glycol)ZnO/Ag	**Film casting method** The created nanocomposites showed improved mechanical characteristics and pH-sensitive swelling behaviour, and they were successfully used as a material for wound dressings.	[117]
4	AgNP-Calcium alginate beads in gelatin scaffolds	**Freeze-drying method** Due to their favourable swelling qualities and non-toxic behaviour against human dermal fibroblasts, they are recommended as acceptable wound dressings.	[118]
5	Chitosan-hyaluronan nano composite sponges	**Ionic cross-linking followed by freeze drying** The material had adequate porosity for applications involving wound healing, good biodegradation, and improved swelling properties.	[119]
6	Methoxy poly (ethylene glycol)-graft-chitosan composite film	**Casting/solvent evaporation method** The substance that was manufactured showed that the medication curcumin had been loaded successfully. The film had an uneven surface without any pores. The produced film has a significant deal of potential for use in wound healing applications, according to an in vitro cytotoxicity research, antioxidant effectiveness assessments, and animal trials (histological study).	[120]
7	Tannic acid/chitosan/pullulan composite nanofibers	**Force spinning method** It has the potential to be used in the treatment of intricate and deep wounds since it replicates a 3D environment, exhibits good water absorption, and encourages fibroblast cell adhesion.	[121]
8	Ag/ZnO nanocomposites	**Deposition precipitation method** The porosity of composites, which ranged from 81 to 88%, the swelling ratios, which ranged from 21 to 24, and the moisture retention period, which ranged from 13 to 14 days, all demonstrated good results in various experiments. These characteristics are all crucial for expediting wound healing.	[122]
9	Silver/hyaluronan bio-nanocomposite fabrics	**Wet-dry-spinning technique** According to in vivo research, fabrics improved the material’s mechanical qualities and increased wound healing effectiveness.	[123]
10	Chitosan-Ag/ZnO composite dressing	**Lyophilisation and immersion method** In many tests, composites performed well in terms of porosity (81–88%), swelling ratios (21–24%), and moisture retention period (13–14 days), all of which are critical elements in improving wound healing.	[124]
11	Starch-AgNPs	**Nanoprecipitation method** By using an ecologically friendly process, alkali-dissolved starch served as a reducing and stabilising agent to create AgNPs, and this strategy may be used for applications in the treatment of wounds.	[125]
12	Cellulose/Polypyrrole/AgNPs/ Ionic liquid composite films	**Simple chemical polymerization method** Composite films demonstrated effective antibacterial action and may be applied as patches to help heal wounds.	[126]
13	Fibrin nanoconstructs	**Water-in-oil emulsification diffusion technique** It served as a reliable carrier molecule for tacrolimus, an immunosuppressant.	[127]

## 9. Advantages of Silver and Fibre Platforms

The ideal wound dressing should fulfill a number of criteria, including those listed below: (i) representing a physical barrier that is permeable to oxygen while also maintaining or providing a moist environment; (ii) being sterile, non-toxic, and protective against microorganism infections; (iii) providing an appropriate tissue temperature to favour epidermal migration and promote angiogenesis; and (iv) being non-adherent to prevent traumatic removal after healing. An ideal wound dressing should have all the qualities listed above, but it is challenging for one kind of dressing to meet every one of these needs. Creating a moist wound environment reduces dehydration and cell death. It facilitates angiogenesis and epidermal migration. It preserves moisture at the site of the wound. Excess exudate must be removed for the wound to heal, but it can also cause healthy tissue to macerate, creating a persistent wound. It enables gaseous exchange. Oxygenation regulates exudate levels and promotes fibroblast and epithelial growth. It prevents infection by prolonging the inflammatory phase and preventing epidermal migration and collagen formation; microbial infections slow the healing of wounds. Low adherence and painless removal of adherent dressings can be uncomfortable and can worsen existing granulation tissue damage. The cost-effective and optimal dressing should promote wound healing while remaining reasonably priced [128,129,130]. The main categories of wound-dressing materials are fibers, gels, membranes, films, sponges, and hydrocolloids, as shown in (Table 2). Nanofiber mats are a superior option for drug delivery compared to all other biomaterials because of their numerous advantages and inherent properties, as shown in (Table 2). The incorporation of nanoparticles into nanofiber scaffolds constitutes a novel approach to “nanoparticle dressing” that has acquired significant popularity for wound healing (Table 2). Due to their remarkable antibacterial capabilities, silver nanoparticles are attractive materials for wound healing. Numerous wound-dressing materials have been created in this area (Table 3), based either on synthetic or natural polymers.

## 10. Silver Nanoparticles Containing Nanofibers for Wound Healing

Ag nanoparticles can generally be incorporated into polymeric nanofibers in various ways (Table 4). Electrospinning a Ag nanoparticle dispersion in a polymer solution is one method. Such a strategy has already been demonstrated to be problematic since the nanoparticles begin to aggregate and lose their effectiveness [159,160]. However, the process of forming Ag nanoparticles onto polymeric nanofibers appears to be advantageous in terms of maintaining the antibacterial activity of the nanoparticles. Table 5 depicts some of the clinical transformation status of silver nanoparticles. Sol-gel and surface functionalization are two approaches that have shown promising relevance in this respect. Similar to the first procedure, adding a precursor salt to the polymer solution before electrospinning results in uniform surface decorating of the nanofibers by the nanoparticles, followed by hydrothermal treatment of the produced nanocomposite nanofibers [161]. An efficient method to produce polymeric nanofibers embellished with metal nanoparticles is surface functionalization by proteins and other substances, such as polydopamine. The functional polymers, such as poly(acrylonitrile-co-glycidyl methacrylate) (PANGMA), can easily link with inexpensive serum albumin proteins such as bovine serum albumin (BSA), for example, by an amine-epoxy process. When submerged in a metal-containing aqueous dispersion, this biofunctionalized nanofiber system can subsequently collect noble metal nanoparticles via a metal-protein interaction. In this circumstance, the protein changes shape from an alpha helix to a beta sheet (Figure 9a), exposing functional groups that can grab biomolecules, metal nanoparticles, etc. (Figure 9b) [160]. Thus, as seen in Figure 9c, the final nanocomposite nanofibers have nanoparticles evenly coated on their surface. Polydopamine created by the self-polymerization of dopamine in an alkaline environment has also provided intriguing prospects for surface functionalization (Figure 9d) [162]. This work was motivated by mussel adhesion onto various surfaces in nature. Through the formation of metal nanoparticles such as Ag, this coating material can diminish metal cations. The long-lasting antibacterial action of the as-synthesized Ag nanoparticles is made possible by their insensitivity to oxidation. The problematic control of coating thickness and surface shape is one of the main problems with polydopamine-based coatings. This flaw prevents the Ag nanoparticles from being distributed uniformly because it causes the surface to become rougher due to the unwanted aggregation of polydopamine in random places [163]. In order to overcome this issue, nanofibers made of poly(dopamine methacrylamide-co-methyl methacrylate), a copolymer inspired by mussels, were electrospun. As a result, Ag nanoparticles can be formed on the surface of these catalytic nanofibers [13].

The silver loaded nanofiber network has shown notable characteristics in wound healing applications (Figure 7 and Figure 8). There are increasing studies on using AgNPs in wound dressings, and all outcomes promoting wound healing are positive [165,166,167]. C. He et al. prepared antibacterial wound dressings whose outermost layer was hydrophobic, which inhibited external microbe adhesion and invasion. Invading microorganisms could be hampered by a specially engineered intermediate region with a high concentration of AgNPs. The polycaprolactone and gelatin hydrophilic surface was employed as the inner layer adhered to the wound bed. The antibacterial activity of the nanofibers containing 10% silver nanoparticles was much higher than that of the dressings containing 1% and 5% silver nanoparticles (Figure 10a). Furthermore, the biocompatibility was far better than that of commercial silver sulfadiazine. This layer also worked as a screen to prevent cytotoxicity from the abundant discharge of AgNPs in the wound’s immediate vicinity. According to the results, the wound dressing’s sandwich structure provides good antibacterial activity and cytocompatibility. The findings pave the path for developing more clinically-relevant wound closure and healing dressings (Figure 10b) [168] AgNPs were evenly distributed throughout the fibers. The XRD patterns of AgNPs and PCL/Gel-AgNPs are depicted in Figure 11. The characteristic peaks at approximately 38° and 44° corresponded to the (1 1 1) and (2 0 0) planes of AgNPs, respectively, proving that AgNPs were successfully doped.

Dalong Li et al. used a combination of sol-gel processing and electrospinning to create Ag-doped mesoporous silica fibers. The main precursors used to make composite fibers were tetraethylorthosilicate, polyvinyl alcohol, and silver nitrate. Water-soluble PVA dispersed silver nitrate better than other polymers used in the preparation of silica composite fiber. After that, the electrospun fibers were heat treated to produce AgNPs via AgNO3 reduction and pyrolyze the PVA component. XPS was performed on the composite nanofiber and pure silica fiber to further investigate the chemical states of the Ag species in the composite nanofiber. Compared to the XPS spectrum of pure silica fiber, composite nanofiber exhibited new peaks attributed to Ag3d (Figure 12A). The typical fully-scanned spectra revealed the presence of Ag, Si, O, and C in the AgNPs/silica composite nanofiber. The C1 peak binding energy of 284.6 eV is used as the calibration standard. The two peaks at binding energies of 368.5 and 374.5 eV, assigned to Ag0 3d5/2 and Ag0 3d3/2, respectively, in the high-resolution XPS spectra (Figure 12B) demonstrate the metallic nature of AgNPs [169].

Suxia Ren et al. created nanofibers with high activity in surface-enhanced Raman scattering (SERS) by electrospinning precursor suspensions of polyacrylonitrile, silver nanoparticles, silicon nanoparticles, and cellulose nanocrystals. Figure 13 displays the FTIR spectra of the electrospun nanostructures. For pure PAN nanofibers, separate peaks at 2244, 1451, and 1096 cm^−1^ correlated with, respectively, the skeletal vibration of the PAN chemical chain, CH wagging vibration, CN stretching vibration, and CH2 bending vibration. The O-H bending of adsorbed water is thought to be responsible for the peaks of about 1658 cm^−1^. The aliphatic CH group vibrations of CH2 are attributed to the peaks at 1340 and 1380. The spectra of PAN/CNC/Ag and PAN/CNC/Ag/Si nanofibers, when compared to those of pure PAN, exhibit new peaks at 1034 cm^−1^ and 824 cm^−1^, respectively, which correspond to the characteristic absorption bands O-H of CNCs and C-H rock, respectively, indicating the presence of CNC in the electrospun nanofibers. It is possible that there were no chemical connections or interactions between the -CN- groups in the PAN and AgNPs because there was no detectable -CN- bond vibration shift [170].

M.R. EI-Aassar et al. constructed a nanofiber scaffold composed of naturally bioabsorbable components, such as hyaluronic and polygalacturonic acid, and embedded with silver nanoparticles for use in vivo. In the interim, silver nanoparticles in this formulation will function as an antioxidant, as antibacterial, and as anti-inflammatory, protecting cells from the harmful effects of high ROS and accelerating wound healing. The Ag nanoparticle of the nanofiber membrane exhibited strong antibacterial zone inhibition efficacy against Gram (+) and Gram (−) bacteria. The increased hydrophilicity and strain activities were the result of the hyaluronic acid component. In addition, the in vivo study on albino rats demonstrated that wound epithelization and collagen deposition were at their highest 14 days following nanofiber delivery. Therefore, it is essential to provide effective nanofibers to heal infected wounds rapidly [171].

J. Shao et al. produced nanofiber membranes comprised of chitosan and silver nanoparticles. In vitro, membranes containing AgNP exhibited antibacterial activities commensurate with the silver release profile. The fibrous membranes with different amounts of AgNPs were fabricated by electrospinning, and SEM and TEM characterized their morphologies. SEM images (Figure 14A–C) showed a uniform fibrous structure for all groups with a similar average fiber diameter of ~200 nm (*p* > 0.05). The TEM image (Figure 14D) demonstrated that the electron-dense AgNPs were formed within the fibers. Figure 14E shows the morphological changes after immersion in PBS or FBS. The membranes kept their fibrous structure after immersion in PBS, while more and more white spots appeared on the fibers with the increase in silver amount and immersion time. Further identification using EDS revealed that the white spots had a high amount of silver and chloride. For the FBS-immersed membranes, a thick layer of protein was observed on the fibrous membranes since day 4. The study utilized an intra-operative contamination model using rat subcutaneous tissue to further evaluate antibacterial efficacy in vivo. The capacity for wound healing was examined using a rat excisional wound splinting model. Researcher findings indicated that AgNP inclusion can improve the antibacterial activity of biomaterials without impairing wound healing.

H.S. Sofi et al. developed wound dressing made of polyurethane incorporated lavender oil and AgNP. Compositional nanofiber dressings offer much potential for use as multipurpose wound dressings since they can both protect from external irritants and promote tissue regeneration. Through diffusion and penetration, lavender oil lowered the stiffness of polyurethane fibers and changed their hydrophobicity. These composite nanofibers dramatically improved the growth and proliferation of CEFs. Studies on cell fixation have shown that fibroblasts grew on fiber mats containing lavender oil and AgNPs in their natural structure [172].

N. Eghbalifam et al. aimed to fabricate a Gum Arabic-based electrospun nanofiber membrane with the ideal porosity, water absorption, and water vapor permeability. It was demonstrated that mat has antibacterial activity against *E. coli*, *Candida albicans*, *S. aureus*, and *P. aeruginosa*. AgNPs nanofibers were coated with PCL using a stable and electrospinnable solution by combining Gum Arabic and PVA. The PCL-coated mat exhibited a high-water absorption capacity and was water resistant. Gram (+) and Gram (−) bacteria and a fungal strain were all suppressed from growing on the composite nanofiber [173].

B. Salesa et al. prepared Carbon nanofibers that are one-dimensional nanomaterials with superior physical and broad-spectrum antibacterial characteristics resistant to antimicrobials. AgNPs are already used in various industrial applications. In HaCaT cells, AgNPs and carbon nanofibers were tested for cytotoxicity, proliferation, and gene expression. AgNPs are smaller and have a completely different morphology than filamentous carbon nanofibers. From pH 5–12, AgNPs displayed a more substantial negative zeta potential than CNFs and equal time-dependent cytotoxicity. These findings hold great potential since they allow AgNPs to be used in various tissue engineering and wound healing applications [174].

G. Sandri et al. aimed to create an electrospun chitosan and glycosaminoglycan membrane incorporated with AgNPs to control bacterial infections during wound healing. The heating method crosslinked the scaffolds to create water-resistant structures, which may also be termed a sterilization process. The systems’ preliminary enzymatic breakdown was tested using lysozyme, which is generally secreted by white cells (macrophages and neutrophils during the inflammatory phase). Nanofibers with a regular uniform shape, a smooth surface, and AgNPs implanted into the polymeric mat creating the fibers were all characteristics of AgNP scaffolds. Antimicrobial activity, primarily against *S. aureus*, was a feature of these systems. The performance of the membrane makes it a viable tool for treating persistent skin lesions [175].

M. Mohseni et al. prepared PCL and PVA nanofibers loaded with various quantities of silver sulfadiazine (SSD) and silver nanoparticles. Wound dressings have sufficient flexibility and hydrophilicity, resulting in satisfactory wound closure adhesion. Silver sulfadiazine is one of the most common antibacterial medicines in wound healing. Antibacterial mats are resistant to *S. aureus*, and increasing the concentration of silver sulfadiazine or AgNPs improves action. The flexibility and hydrophilicity of PCL and PVA nanofibers allowed the wound closure to be moistened with wound fluid loaded with cytokines and growth factors. On the other hand, SSD was more hazardous to fibroblast cells than AgNPs [176].

Z.W. Li et al. synthesized polyvinyl alcohol, Chitosan oligosaccharides, and AgNP nanofibers to stimulate HSF adhesion, proliferation, and cell cycle transition from the quiescent G0/G1 phase to the active S DNA synthesis and the active G2/M phase of division. Using RT-PCR, immunofluorescent labelling, and Western blot analysis, investigators demonstrated that the nanofibers upregulated molecules implicated in the TGF-β1/Smad signaling pathway, hence boosting collagen synthesis and improving wound healing by upregulating TGF-β1 secretion and activating the TGF-β1/Smad signaling pathway during the early stages of wound healing, boosting the adhesion and proliferation of fibroblasts. This occurrence is followed by an acceleration in the proliferation and differentiation of keratinocytes, an increase in the synthesis of collagen and the extracellular matrix, the facilitation of granulation tissue development and angiogenesis, and finally, the promotion of re-epithelialization. This research offers a significant step forward in creating novel, enhanced drug delivery vehicles for therapeutic wound-healing medicines [177].

M. Mostafa et al. used an electrospinning approach to manufacture AgNPs loaded in polystyrene nanofiber scaffold. In biomedical fields, this can be used as a potent bioactive substance. Furthermore, AgNPs-polystyrene nanofibers have improved their antibacterial effect against both *S. aureus* and *E. coli*. Various electrospinning settings were investigated, with this nanofiber proving the most effective. DMF can be used as a reducing agent to make Ag nanoparticles with a diameter of 21–40 nm [178].

E. Esmaeili et al. designed nanofibrous scaffolds from polyurethane and cellulose acetate. Due to their significant antibacterial action, reduced graphene oxide/silver was also employed in the mats. By coming into direct contact with bacteria, scaffolds could inhibit them. Curcumin has the most significant impact on wound healing and can speed up the healing of artificial wounds. Curcumin and graphene oxide/silver nanocomposites can be combined to create ultrafine, bead-free nanofibers with a porous structure, making them suitable for biomedical applications. On direct contact with microbial cells, their antibacterial activities result in a 100% inactivation rate for Pseudomonas bacteria and a 95% inactivation rate for *S. aureus* bacteria. In this study, in vivo histopathological studies revealed that adding curcumin can considerably improve wound healing and epidermal layer regeneration [179].

H. Bardania et al. synthesized silver nanoparticles biogenically produced using Teucrium polium extract and implanted in PLA and PEG mat to offer an absorbable wound dressing with antibacterial and antioxidant properties. Antibacterial tests revealed that *S. aureus* and *P. aeruginosa* were sensitive to biosynthesized AgNPs at different concentrations and had a good safety profile in human macrophage cells. This green synthesis method has proven to be a quick, cost-effective, and efficient way to make AgNPs without needing external stabilizers or reducing agents [180].

S.M. Hong et al. developed antibacterial polyurethane nanofiber textiles with Ag nanoparticles based on polycarbonate diol/isosorbide. The materials were pliable, with breaking strains ranging from 355 percent to 950 percent under a tensile stress of 7.28 to 23.1 MPa. Cell proliferation was performed using the HaCaT cell line, which exhibited cytocompatibility and no toxicity. Researchers also tested antimicrobial capabilities against *S. aureus* and methicillin-resistant MRSA. Adding AgNPs to the polyurethane nanofiber membrane enhances physicochemical parameters such as mechanical, thermal, and biological properties [181].

M.A. Mohamady et al. aimed to create core-shell electrospun membranes that could promote cell proliferation and act as antibacterial agents. A polycaprolactone shell membrane was used to load phenytoin. Silver-chitosan nanoparticles were placed in a PVA core layer as biocidal agents. Researchers employed coaxial techniques to create dual-drug delivery systems suitable for the model pharmaceuticals. The use of phenytoin to improve the cytocompatibility of scaffolds was proposed. In the core, biocides such as CS-coated silver nanoparticles were used as biocides, while the shell contained phenytoin, a healing-promoting substance [182].

R. Liu et al. designed a novel hydrogel dressing created by crosslinking cellulose fibers with gelatin and aminated AgNPs (Ag-NH_2_ NPs). The use of multiple components significantly enhanced the mechanical, self-recovery, hemostatic (gelation), antibacterial, and fluid balancing capabilities of the wound bed. In vitro and in vivo assessment of the mat wound healing model revealed excellent biocompatibility and healing performance (90 percent healed and 83.3 percent survival after 14 days) [183].

M. Hu et al. synthesized electrospun nanofibers of ZnO, Ag, PVP, and PCL. ZnONPs had an average diameter of nanofibers of 40.07 ± 9.70 nm and AgNPs’ diameter of 37.46 ± 12.02 nm, respectively. The antibacterial effects of the single metal material embedded scaffold were far superior to those of the single metal material-loaded nanofibers against *S. aureus* and *E. coli*. Adding ZnO and Ag to these nanofibers lowered their cytotoxicity against fibroblasts [184].

J.P. Ye et al. created soluble keratin from wool using keratinase and raised the molecular weight of the keratin to 120 kDa using TGase. In situ bio-reduction was used to create Ag nanoparticles on nanofibers using the keratinase stated earlier as the reducing agent. High molecular weight keratin had improved mechanical and hydrophilic qualities when co-electrospun with Poly(3-hydroxybutyrate-co-3-hydroxy valerate). *E. coli* and *S. aureus* were utilized to test the antibacterial effectiveness. The researcher demonstrated that they both possessed potent antibacterial properties. The mats dramatically accelerated the healing of skin wounds [185].

M.K. Ahmed et al. designed the doped antimicrobial silver nanoparticles incorporated into nanofibrous polymeric scaffolds. Ag concentration in the scaffolds’ magnetite phase increased the human melanocyte survival and antibacterial activity against *E. coli* and *S. aureus*, preventing dermal and epidermal abnormalities on day 10. In rats, Ag exposure accelerated wound healing. Large pores in this porous structure capture nanoparticles, possibly increasing the material’s properties. Morphological studies show that Ag reduces nanoparticle agglomerates. All compositions were highly biocompatible when tested against a human skin melanocyte, with viability rising as Ag ions in their magnetite phase increased [186].

C. Tonda-Turo et al. prepared antibacterial medication gentamicin sulfate or silver nanoparticles placed into gelatin nanofibrous matrices in this study to elicit a significant antibacterial activity against Gram (+) and Gram (−) microorganisms. Only water was employed as a solvent throughout the process, resulting in gelatin-crosslinked nanofibers doped with antibacterial compounds in an environmentally and cell-friendly approach to green electrospinning. Because of their dual qualities of tissue development support and antibacterial capabilities, the discovered matrices are potential membranes for use in wound healing. Biomolecules (e.g., growth factors) capable of aiding the regeneration process will be loaded into them with a minimum risk of denaturation, the green electrospun approach [187].

M. Bagheri et al. designed silver and zinc oxide nanoparticles embedded in chitosan nanofibers. Nanocomposites’ antibacterial and antioxidant activities against *E. coli*, *S. aureus*, and *P. aeruginosa* were discovered. After 24 h of treatment, the scratches significantly improved. According to the impact of coagulation time, the nano-scaffold also exhibited good blood compatibility. In the wound-healing experiment, there was also a lot of movement and growth of fibroblasts at the wound edge [188].

A.M. Abdel Mohsen et al. synthesized hyaluronan/silver bio-nano composite textiles are used to create a novel material in which Ag-NPs are generated in situ. In-situ produced hyaluronan and AgNPs were used to prepare fibers for the first time. Fibers were tested against *E. coli* K12 and were found to have significant bactericidal activity. The fibers have no cytotoxicity when tested against a HaCaT. The in vivo study revealed that the produced fibers have a high healing efficacy and significantly speed up the healing process.

R. Ashraf et al. investigated a unique technique for fabricating nanofiber scaffolds made of cellulose assimilated with TiO_2_ and AgNPs. The biocompatibility and bioactivity of the produced nanofibers were determined using cell viability experiments using chicken embryo fibroblasts. Model microorganisms were used to evaluate the antibacterial characteristics of these scaffolds (e.g., *E. coli* and *S. aureus*). Electrospun nanofibers are an effective scaffold for cell proliferation. The antibacterial ability of the scaffold was conferred by in situ adsorption of AgNPs, as evidenced by disc diffusion techniques’ suppression of bacterial strains [189].

A.J. Hassiba et al. synthesized a double-layered nanocomposite nanofibrous mat made of an upper layer of PVA and chitosan loaded with AgNPs, and a bottom layer of chlorhexidine was created by electrospinning. According to thermal measurements, the PVP-drug-loaded layer showed maximum thermal stability, which justifies future investigation for various wound-healing applications. The in vitro study obtained tests against *E. coli*, *S. aureus*, *C. albicans*, and *P. aeruginosa*. In this work, researchers created two brand-new, dual-purpose nanofiber wound dressings. The identical system used two electrospun layers loaded with the same medication in both dressing designs. A barrier against microbial invasion might infect the wound into the top layer. Its ability to combat microbes results from AgNPs’ antibacterial activity [190].

Q.H. Tran et al. fabricated PU/Ag nanocomposite dressings that provide broad-spectrum antimicrobial efficacy against microorganisms that are easy to use and offer a warm, moist healing environment that protects and cleanses a wound while promoting the body’s natural healing process [191].

S. Amer et al. developed biocompatible electrospun binary nanofiber mats that were created by combining PVA and gelatin. The silver nanoparticles were added to the PVA and gelatin mixture. Before being used as wound dressings, both types of mats were studied using SEM evaluation. The researchers studied both kinds of wound healing, and both enhanced the microscopic quality of the healed skin, though not at a faster rate. Both membranes with and without AgNPs successfully combated microbial invasion into the wound bed. Both mats increased the quality of the restored skin [192].

D. Yan et al. was the first attempt at creating optimal ocular bandages with the ability to promote cell growth while also being antibacterial. It has the potential to rescue patients from the problems caused by the repeated application of antibiotics to the skin to prevent infections. Fungal keratitis is one of the most common causes of blindness, and typical medical treatment is ineffective. In this study, investigators produced surface-modified PLA electrospun fibers with AgNPs and cellulose nanofibrils for cell proliferation and antimicrobial application. The scaffold had a surface modification that has excellent biocompatibility and antibacterial characteristics. The application of membrane to their surface wettability, which caused the value of WCA to drop substantially from 130.4 to 0°, increased cell proliferation, which was necessary for wound healing [193].

G. Rath et al. reported silver nanoparticle-composite collagen nanofibers, and the histological investigation revealed increased collagen formation, re-epithelization, and better wound contraction. These findings were attributed to silver nanoparticles’ anti-inflammatory and aseptic properties, resulting in less reactive cell infiltration and encouraging fibrous connective tissue proliferation and successive keratin layer regeneration [194].

**Table 4 polymers-14-03994-t004:** Incorporation of silver nanoparticles into electrospun nanofibers for wound healing.

S. No.	AgNPs with Polymers	Solvents	Voltage, Distance, Flow Rate			Diameter (nm)	Antibacterial Efficiency (ZOI (mm), MIC or %)	Bio- Compatability	Ref.
			(kV)	(cm)	(mL/h)				
1	PLA	Methylene chloride, DMF	14	10	3	1.44 ± 0.32 μm	*S. aureus*—6.5 mm *P. aeruginosa*—9.3 mm	CjECs CECs	[85]
2	PCL-Gelatin	Acetone	15	15	1	830–920	*E. coli*—1.53 ± 0.32 mm	HDF	[168]
3	Polyurethane	THF	15	15	0.5	200–2000	*E. coli*—16.2 ± 0.8 mm *S. aureus*—8.7 ± 1.2 mm	CEFs	[172]
4	Gum Arabic-PVA-PCL	DMF DI-water	18	15	0.5	150–250	*E. coli*—2.5 mm *S. aureus*—2.9 mm	MEF	[173]
5	PCL-PVA	CHCl_3_, CH_3_OH, H_2_O	27	-	3	-	*S. aureus*—90 mm	HDF	[176]
6	Polystyrene	DMF	2–3	-	1–3	96–471	*E. coli*—11 mm *S. aureus*—4.0 mm	-	[178]
7	Polyurethane	HFP	17	20	1.5	500	*S. aureus*—20.41 mm MRSA—18.24 mm	HaCaT	[181]
8	PVA-PCL	CHCl__3__, CH3OH, Water	23	15	0.02	70 nm	*E. coli*—14 mm *S. aureus*—18 mm	NIH3T3	[182]
9	PCL	water	18	16	1	0.38 μm	*S. aureus*—79.2 ± 4.5 % *E. coli*—80.1 ± 4.9%	HFB4	[186]
10	PVA-TPU	Water: DMF	25	10	1	230–280	*S. aureus*—50 μg/mL *E. coli*—25 μg/ml	-	[188]
11	Chitosan- PEO	Acetone	20	14	-	100–300	*E. coli*—20 ± 2 nm	HDF	[123]
12	PCL-Cellulose acetate	Acetone: DCM	22	16	1	2–6.3 μm	*S. aureus*—18 mm *P. aeruginosa*—10 mm	HOB, HFB4	[190]
13	Collagen	HFIP	18	10	25 mL/min	300–700	*S. aureus*—3.2 cm *P. aeruginosa*—2.3 cm	No toxic on rat skin	[194]
14	PLA-PVP	DCM	20–30	15	2	500–650	*E. coli*—96.7 % *S. aureus*—96.9 %	–	[195]
15	Cellulose acetate-PVAc	Acetone water	25	10	0.8	1.33 ± 0.63µm	*S. aureus*—9.2 ± 1.6 mm *E. coli*—8.2 ± 0.9 mm	CEFs	[196]

**Table 5 polymers-14-03994-t005:** Clinical transformation status of some silver nanoparticles-based products.

S. No.	Study Title (ClinicalTrial Identifier ID)	Status of Clinical Trails
1	Topical Application of Silver Nanoparticles and Oral Pathogens in Ill Patients (NCT02761525)	Completed
2	Topical Silver Nanoparticles for Microbial Activity (NCT03752424)	Unknown
3	Silver Nanoparticles in Multidrug-Resistant Bacteria (NCT04431440)	Completed
4	Efficacy of Silver Nanoparticle Gel Versus a Common Antibacterial Hand Gel (NCT00659204)	Unknown
5	P11-4 and Nanosilver Fluoride Varnish in Treatment of White Spot Carious Lesions (NCT04929509)	Recruiting
6	Evaluation of Diabetic Foot Wound Healing Using Hydrogel/ Nano Silver-based Dressing vs. Traditional Dressing (NCT04834245)	Completed

## 11. Conclusions Challenges and Perspective

Wound healing with nanofibrous platforms loaded with silver nanoparticles has been studied biologically in vivo and in vitro as well as mechanically in this review. Because of their unique physicochemical and biological characteristics, AgNPs have drawn significant attention from researchers working on applications for wound healing. Ag nanoparticle-loaded electrospun nanofiber scaffolds also showed exceptional antibacterial activity, high porosity, non-toxicity, and biodegradability. Due to their hydrophilic qualities and prolonged release pattern, these nanofiber scaffolds have become more and more well-liked on a global scale. By promoting and hastening the healing process, they contribute significantly to wound dressing. Additionally, silver nanoparticles and other antibacterial substances together showed synergistic antibacterial properties. The effectiveness of silver nanoparticles in wound healing and skin regeneration has been established in numerous papers from various researchers. Without a doubt, additional research will produce scaffolds with unique properties beneficial for treating chronic wounds. However, since it avoids hazardous chemicals, manufacturing silver nanoparticles using green chemistry is an intelligent strategy. Silver nanoparticles can be produced through green chemistry and used for wound dressings to make them non-toxic and compatible with the body. Much research has been conducted on silver nanoparticles to improve the antibacterial capability of medical products such as wound dressings. Nonwoven mats made of electrospun nanofibers offer a great deal of potential for use in wound healing since they structurally resemble native extracellular matrix.

Various biomedical applications, including leading-edge research aimed at healing chronic diabetic wounds, have investigated potential methods of nanotechnology-based medication delivery. In addition to their nanometric size, AgNPs were discovered to have possible use in treating diabetic wounds to lessen the likelihood of limb amputation. Their excellent antibacterial activity, anti-inflammatory response, and non-toxic nature make them an ideal and suitable alternative to other nanomaterials for wound dressing. AgNPs’ advantageous physicochemical characteristics support antibacterial effectiveness, and their surface charge also enables surface functionalization by coordinating particular ligands on the surface to try target-specific delivery. Consequential research has demonstrated the biocompatible AgNPs’ promise for treating diabetic wounds effectively, and a handful of the compounds have already been approved for commercialization. Several might be available soon with improved efficacy as an optimal dressing for successful wound healing in diabetes patients, according to a concurrent clinical study in human subjects.

Additionally, those investigators found a combination of AgNPs and biopolymers to be more effective, and the inclusion of growth factors or phytochemicals may hasten wound healing by correcting any tissue damage. The exponential rise in research papers on green synthesis, when considering the AgNPs synthesis methods, is a drawback since it demonstrates the value and interest of plant materials in the production process. The synthesis rate of AgNPs is increased by using this economical and ecologically beneficial technique. However, morphological traits play a significant role in how effective AgNPs are. For the AgNPs to have the desired features, such as superior wound healing properties, it is necessary to standardize the optimization of the plant extracts and other reactive product characteristics.

Additional research is required to link the physiological attributes of AgNPs with the physiological milieu in which they act. Before widespread usage in wound care products, careful consideration of their toxicity must be made because silver nanoparticles are incredibly active relative to their bulk. Extensive research into short- and long-term toxicity studies should be necessary to ascertain the underlying mechanism, and it should take thorough in vivo investigations into consideration as one of the future possibilities in developing AgNPs for wound healing. Additionally, special care must be given to the ideal AgNP dosage in formulations and suitable pairings to achieve a superior response in the diabetic wound. In recent years, nanotechnology has enabled the fabrication of various forms of AgNPs. However, AgNPs’ efficacy is hindered by their propensity for aggregation; surface passivator reagents are usually required to avoid accumulation. Further, silver oxidation may produce reactive oxygen species and radicals that can harm intracellular micro-organelles (such as mitochondria, ribosomes, and vacuoles) and macromolecules such as DNA, protein, and lipids. AgNPs are biocompatible but can occasionally result in argyria, according to research on how risk-free they are for people with DFU infections. However, most current studies on electrospun nanofibers throughout wound healing have been limited to pharmacodynamic assessments. As a result, the precise mechanism underlying nanofiber-assisted wound healing is unknown. Despite their increasing applications, comprehensive biological information still requires additional research due to several controversial results published on their safety. Researchers used chemical reduction methods to create a stable and colloidal dispersion of AgNPs using borohydride and hydrazine as reducing agents. Whereas these reduce the agent’s hyperactivity, they are toxic to the environment, limiting their applications.

## Figures and Tables

**Figure 1 polymers-14-03994-f001:**
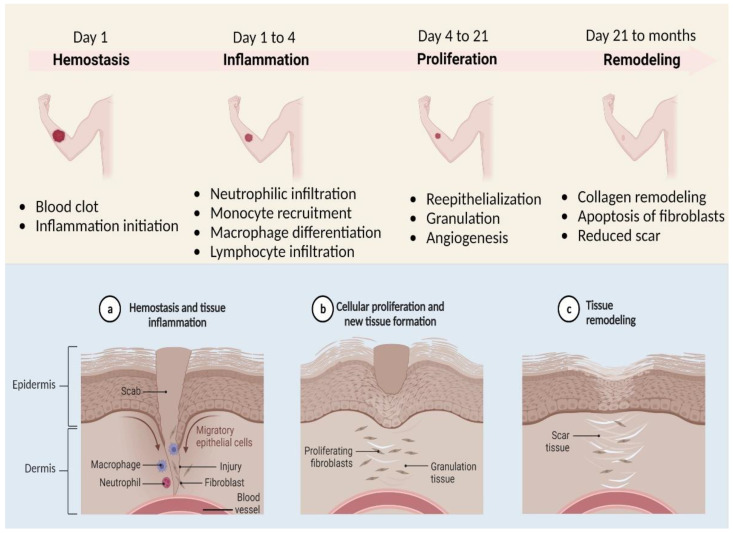
A diagrammatic illustration of the basic steps of cutaneous wound healing.

**Figure 2 polymers-14-03994-f002:**
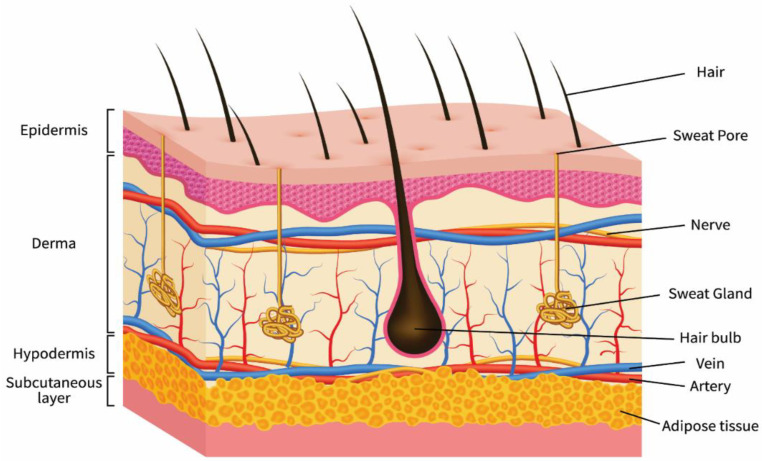
A diagrammatic illustration of a three-dimensional (3D) structure of the skin.

**Figure 3 polymers-14-03994-f003:**
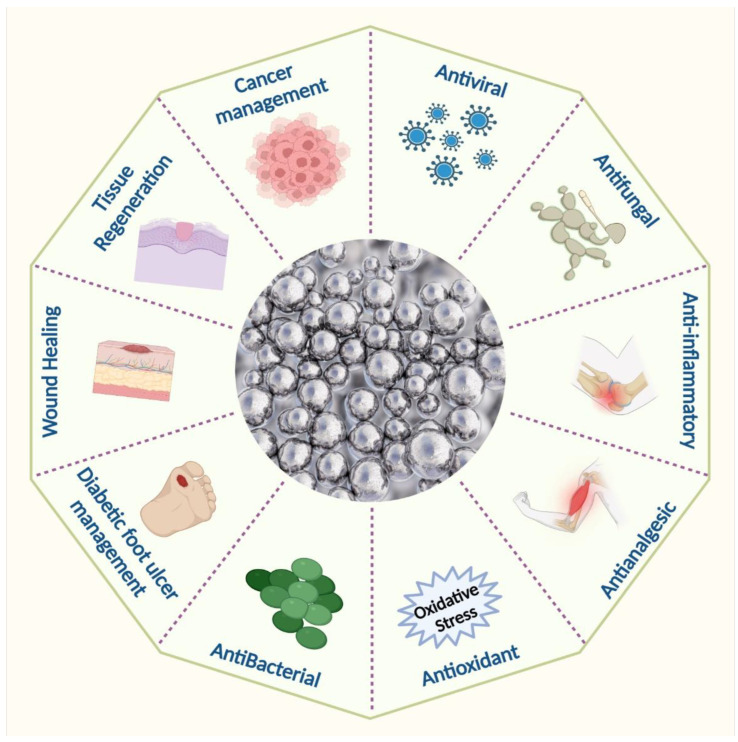
Biomedical applications of silver nanoparticles.

**Figure 4 polymers-14-03994-f004:**
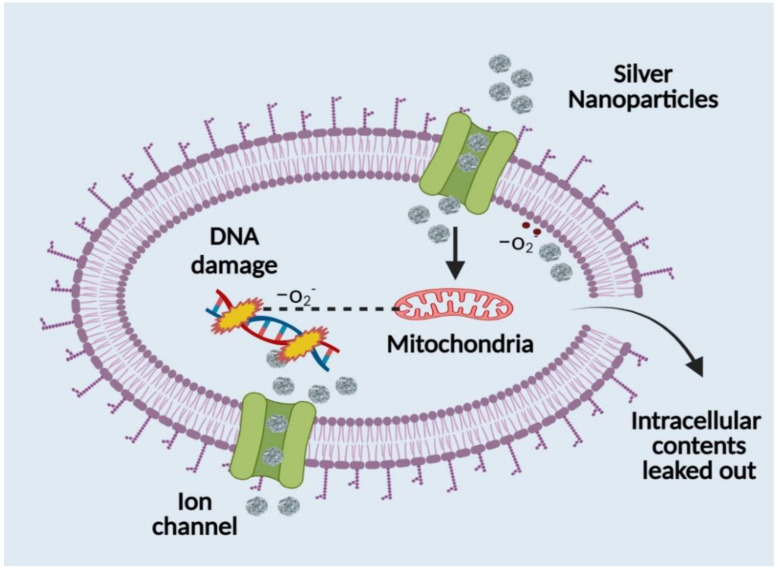
Antibacterial mechanism of silver nanoparticles.

**Figure 5 polymers-14-03994-f005:**
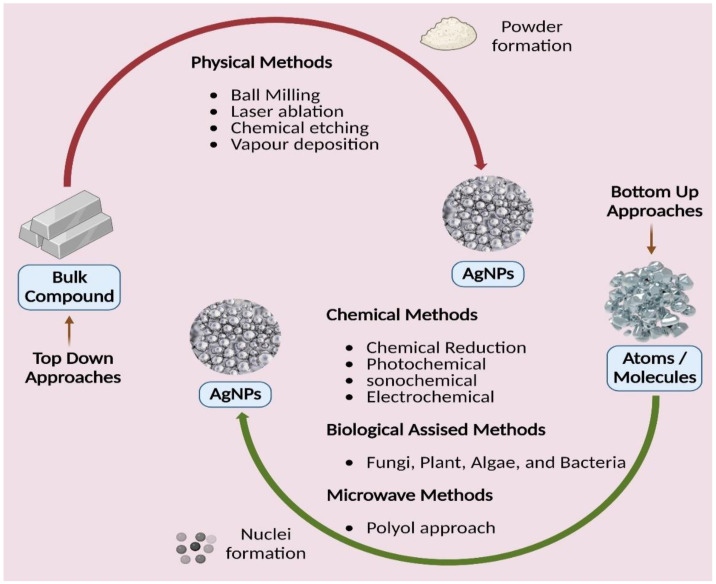
Different approaches of silver nanoparticles synthesis.

**Figure 6 polymers-14-03994-f006:**
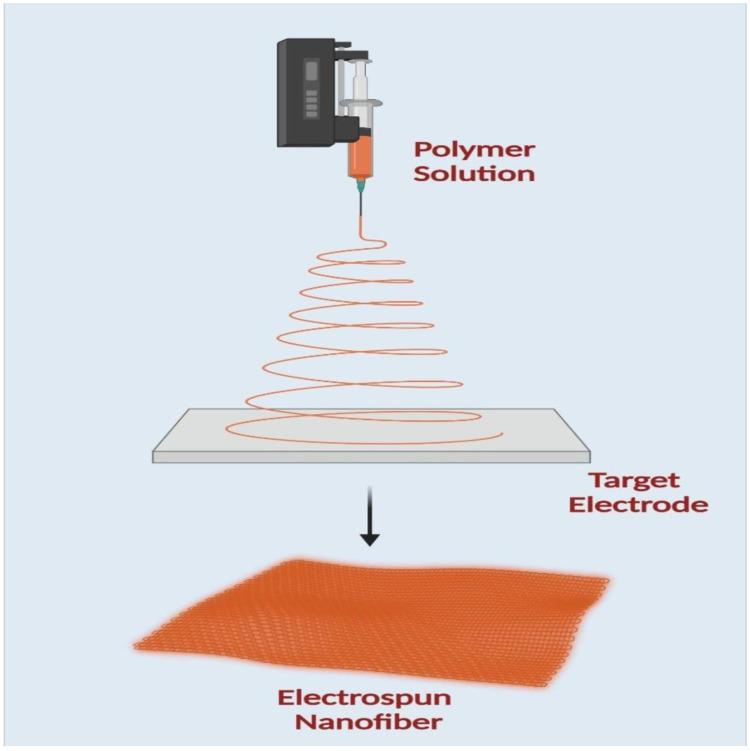
A schematic diagram of electrospinning apparatus.

**Figure 7 polymers-14-03994-f007:**
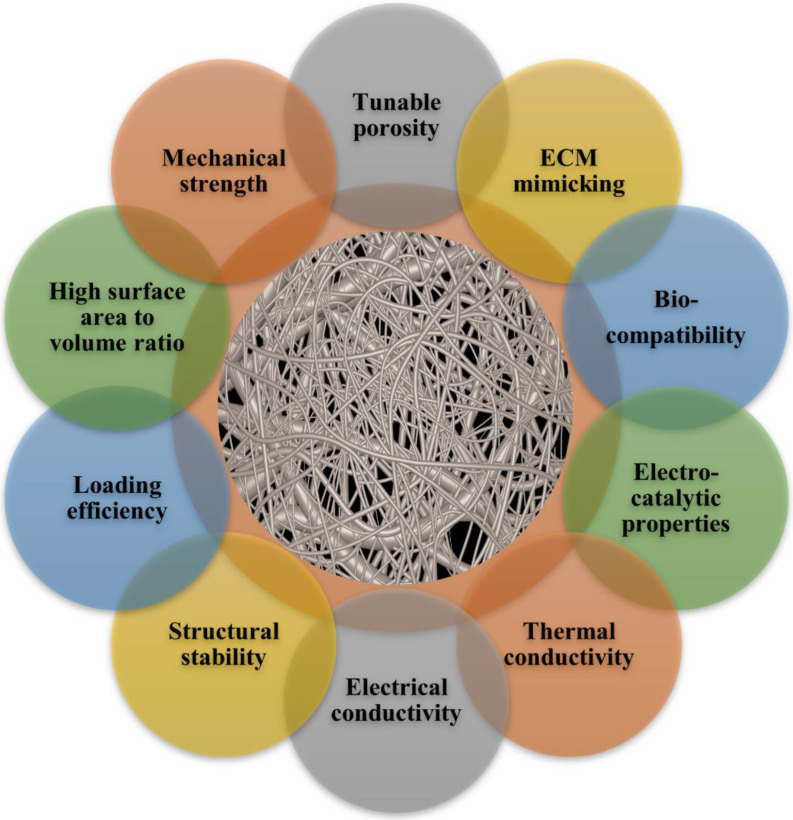
Key properties of electrospun nanofibers.

**Figure 8 polymers-14-03994-f008:**
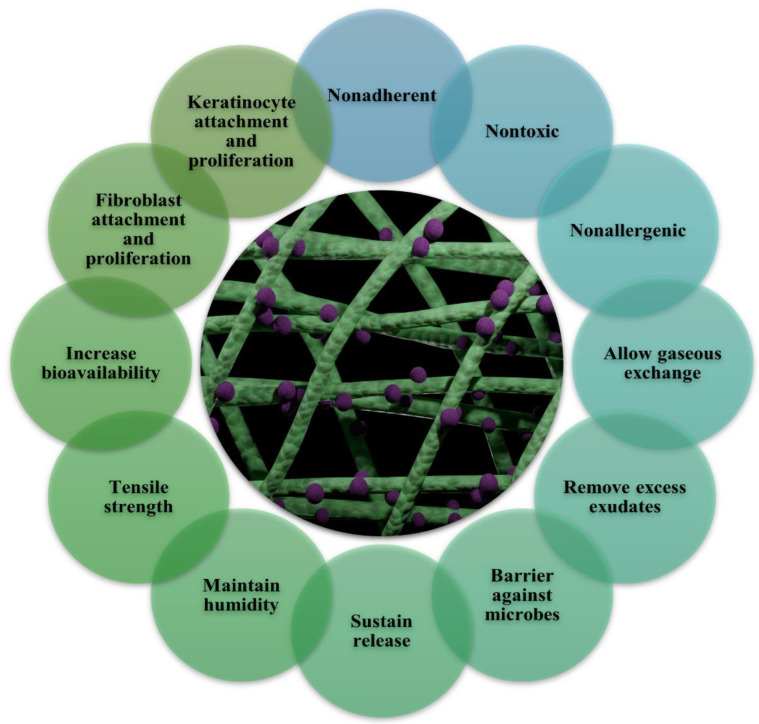
Characteristics of AgNPs containing nanofibers on wound healing.

**Figure 9 polymers-14-03994-f009:**
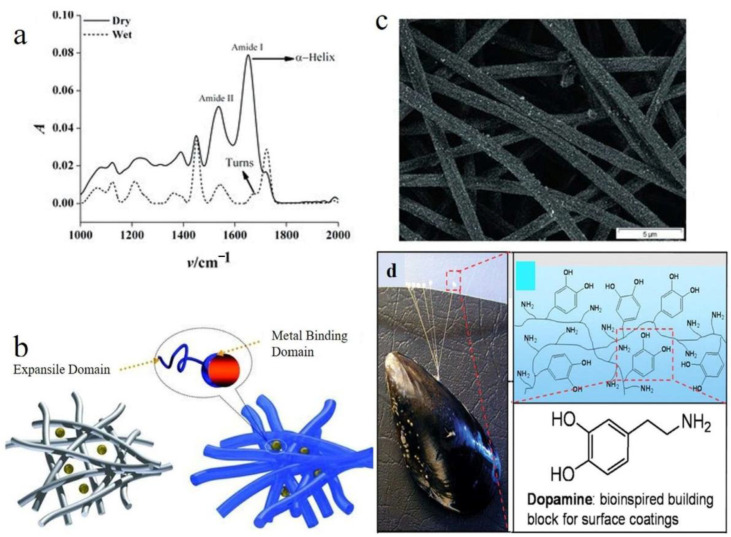
(**a**) ATR-FTIR spectra for the BSA functionalized PANGMA nanofibers in different hydration states (of dry and wet represented by the solid and dashed lines, respectively). (**b**) Schematic illustration of the capturing process of the metal nanoparticle by the swollen functionalized nanofibers). (**c**) SEM image shows uniform distribution of metal nanoparticles across the nanofibers. Reproduced with permission [160]. Copyright 2012, Wiley-VCH. (**d**) Camera image of a mussel stuck onto a polymeric surface along with a simple representation of the amine and catechol groups of the dopamine building block for surface coating. Reproduced with permission [164]. Copyright 2007, Science AAAS.

**Figure 10 polymers-14-03994-f010:**
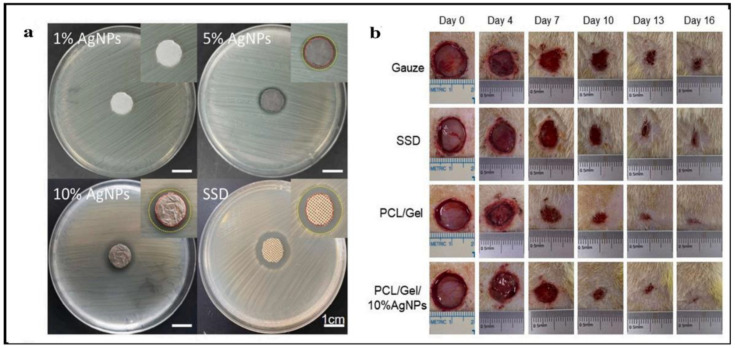
(**a**) Representative photograph showing the antibacterial zones of *E. coli* growth inhibition by the different dressings. (**b**) Photographs show the wound healing process at different time points; images reproduced with permission from [168] ©2021 Materials Science & Engineering C. Published by Elsevier Ltd.

**Figure 11 polymers-14-03994-f011:**
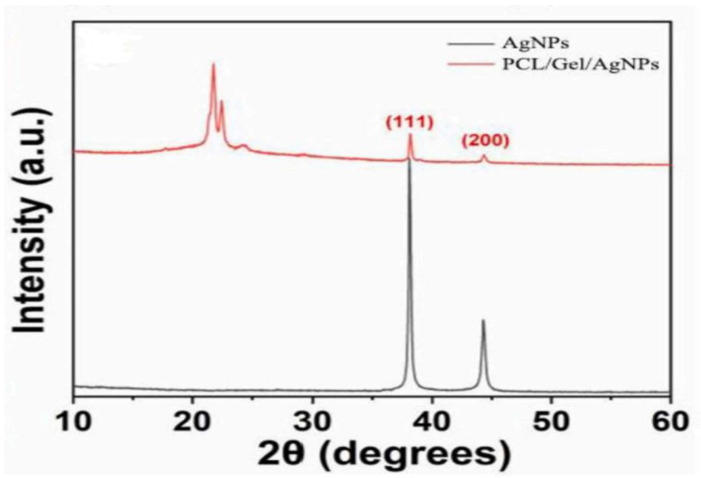
XRD patterns of AgNPs and PCL/Gel-AgNPs. Images reproduced with permission from [168] ©2021 Materials Science & Engineering C. Published by Elsevier Ltd. (Amsterdam, The Netherlands).

**Figure 12 polymers-14-03994-f012:**
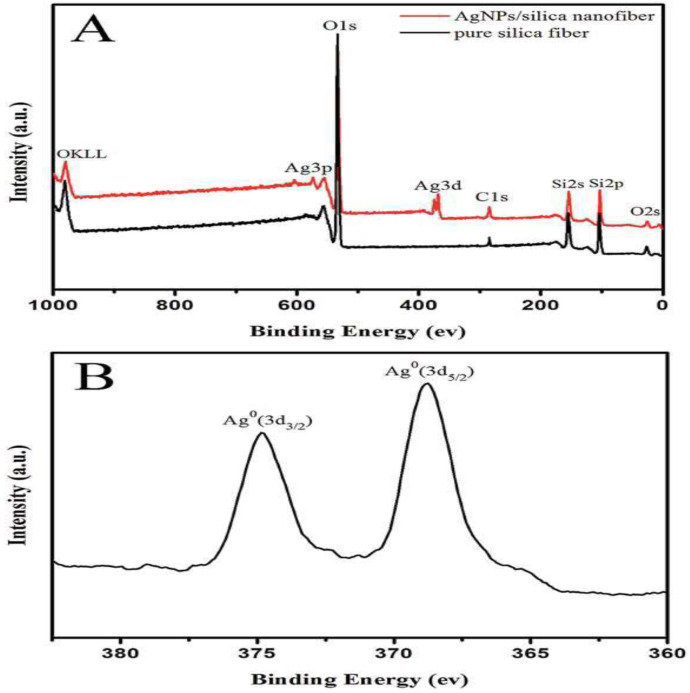
XPS spectra of composite nanofiber: (**A**) full XPS spectrum of AgNPs/silica nanofiber and pure silica nanofiber, (**B**) Ag3d of AgNPs/silica nanofiber. Reproduced with permission [169]. Copyright 2016, Royal Society of Chemistry.

**Figure 13 polymers-14-03994-f013:**
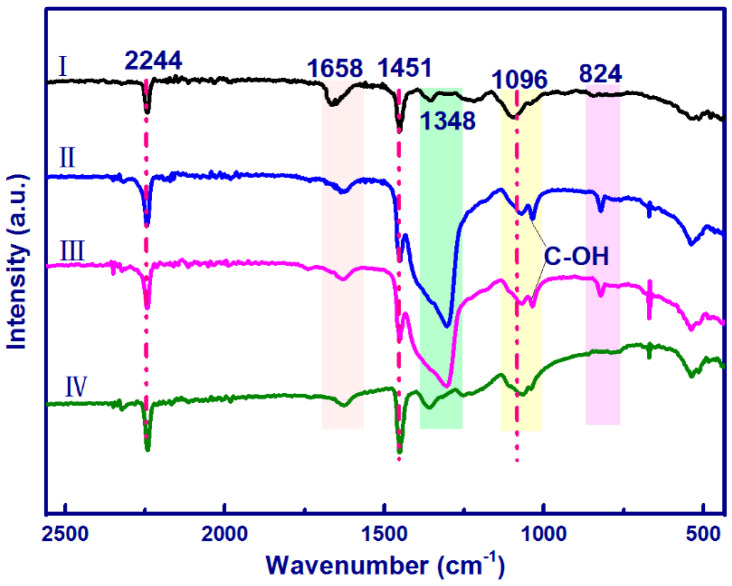
FTIR spectra of the electrospun nanofibers: (**I**) PAN; (**II**) PAN/CNC/Ag; (**III**) PAN/CNC/Ag/Si; and (**IV**) p-PAN/CNC/Ag/Si. Reproduced from [170] Materials (MDPI).

**Figure 14 polymers-14-03994-f014:**
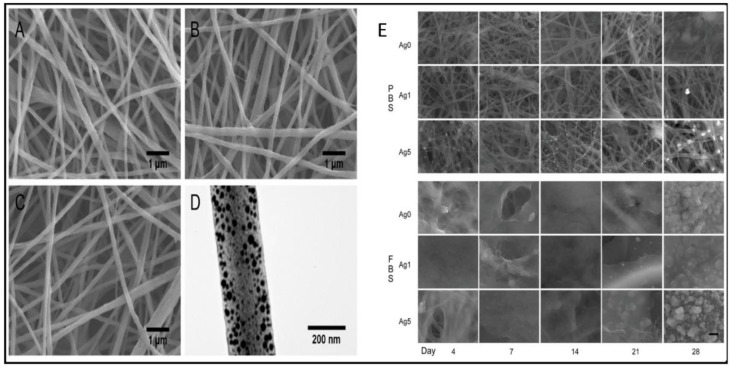
The SEM images of the electrospun membranes of (**A**) Ag0, (**B**) Ag1, and (**C**) Ag5 groups. (**D**) The TEM image of a single fiber from the Ag5 group, where the electron-dense areas were silver nanoparticles. In vitro interaction with PBS and FBS. (**E**) The SEM micro-images of membranes after being immersed in PBS and FBS. Scale bar: 1 μm; images reproduced with permission from [147] ©2019 Materials Science & Engineering C. Published by Elsevier Ltd.

**Table 2 polymers-14-03994-t002:** The benefits and drawbacks of the various types of nanomaterials.

S. No.	Wound Dressing Type	Advantages	Disadvantages	Ref.
1	Fibers	nonadherent, nontoxic, nonallergenic allow gaseous exchange remove excess exudates barrier against microbes sustain release maintain humidity tensile strength increase bioavailability fibroblast attachment and proliferation keratinocyte attachment and proliferation tunable porosity ECM mimicking bio-compatibility electro-catalytic properties thermal conductivity electrical conductivity structural stability loading efficiency high surface area to volume ratio mechanical strength	unsuitable for third degree, eschar, and dry wounds if the wound is highly exudative, need a secondary dressing	[131]
2	Membranes	act as physical barriers membranes simulate extracellular matrix (ECM) structure assure gas exchange, cell proliferation, and nutrient supply	the materials and solvents used in the production process may be harmful	[132]
3	Films	impermeable to bacteria allows the healing process to be monitored painless removal	hard to handle non-absorbent adhere to the wound bed and cause exudate accumulation	[133]
4	Hydrocolloids	non-adherent high density painless removal high absorption properties	can be cytotoxic have an unpleasant odor low mechanical stability maintain acidic pH at the wound site	[134]
5	Hydrogels	high absorption properties provide a moist environment at the wound site water retention oxygen permeability ensure the solubility of growth factor/antimicrobial agents	weak mechanical properties need a secondary dressing	[135]
6	Sponges	high porosity thermal insulation sustain a moist environment absorb wound exudates enhance tissue regeneration	mechanically weak may provoke skin maceration unsuitable for third degree burn treatment or wounds with dry eschar	[136]

**Table 3 polymers-14-03994-t003:** A summary of available AgNP-based wound dressing products and their benefits.

S. No.	Wound Dressing Materials	Size of AgNPs (nm)	Target Microbe	In Vivo/In Vitro Model	Advantage of Nanocoating	Ref.
1	Chitosan-Poly Vinyl Pyrrolidone (PVP) composite	10–30	*E. coli* and *S. aureus*	L929 cell line	Compared to the control sample, silver nanocomposite reduced the amount of inflammatory cells by 99.	[137]
2	Plumbagin caged AgNP-collagen scaffolds	60 nm	*E. coli* and *B. subtilis*	wistar rat/Swiss 3T6	The antibacterial and wound-healing capabilities of silver and plumbagin in the PCSN cross-linked collagen scaffold showed the importance of nano-biotechnology.	[138]
3	Chitosan/Poly (Ethylene Oxide) matrix	5	*E. coli*	-	AgNPs, because of their size and structure, were found to increase antibacterial activity when introduced.	[139]
4	Chitin/nanosilver composite scaffolds	5 nm	*E. coli* and *S. aureus*	L929	The scaffolds are antibacterial and have excellent blood clotting capabilities, which will help with wound healing. These scaffolds were hazardous to mouse fibroblasts in vitro. Whether in vitro cytotoxicity affects in vivo wound healing is unknown.	[140]
5	Activated Carbon coated silver nanocomposite	50–400	*S. aureus*, Klebsiella pneumoniae and *P. aeruginosa*	-	When compared to plain activated carbon, the Ag composites’ antibacterial activity was significantly higher.	[141]
6	Silver nano-coatings on cotton gauzes	100–300 nm	*S. aureus*	HaCaT/3T3	The developed textile materials show promise as an alternative to traditional wound dressings due to their antimicrobial properties and biocompatibility.	[142]
7	Polyurethane Foam mixed Ag-NPs Dressing	100	*E. coli*, *P. aeruginosa* and *S. aureus*	Human fibroblast	Wound healing was enhanced by the use of the foam dressing.	[143]
8	AgNP gelatin hydrogel pads	7.7–10.8 nm	*E. coli*, *S. aureus P. aeruginosa*	Human’s normal skin fibroblasts	Gelatin hydrogel pads infused with silver nanoparticles have shown promise as antibacterial wound dressings.	[144]
9	Chitosan-PEG hydrogel	75	*E. coli*, *P. aeruginosa* and *S. aureus*	Rabbit	On day 14, the dermal layer of skin and the collagen pattern were both healthy in the Ag-NPs impregnated chitosan-PEG hydrogel group.	[145]
10	AgNPs incorporated Pluronic F127 and Pluronic F68 thermosensitive gel	-	*E. coli*, *S. aureus* and *P. aeruginosa*	-	Gel may disrupt the structure of bacterial cell membranes, allowing the substance to enter the cell, where it can condense DNA, combine and coagulate with the cytoplasm, and ultimately kill the bacteria by causing the cytoplasmic component to leak out.	[146]
11	Chitosan nanofiber	25	*S. aureus*	Wistar Hannover rats	Biological media had a substantial impact on the release of silver; proteins blocked the release of the metal, whereas inorganic ions slowed it down. As a result, to elicit in vivo antibacterial activities, a high concentration of AgNPs was required.	[147]
12	Asymmetric Wettable Chitosan nanocomposite	25	*E. coli*, *P. aeruginosa* and *S. aureus*	HEK293 cell line	The dressing has been shown to encourage cell growth in an in vitro cytocompatibility study.	[148]
13	Cellulose hydrogel	5–50	*E. coli* and *S. aureus*	New Zealand rabbit	Three days faster wound healing was seen using nanohydrogel compared to the control group.	[149]
14	Chitosan gels	15	*P. aeruginosa*	Human dermal fibroblasts	Chitosan gels containing AgNPs showed improvement in biocompatibility tests on primary fibroblasts.	[150]
15	Silk fibroin/ carboxymethyl chitosan composite sponge	4.9 ± 1.9 nm	*S. aureus* and *P. aeruginosa*	-	This AgNP-loaded SF/CMC sponge shows promise as a potential antimicrobial wound dressing.	[151]
16	Chitosan cross-linked bilayer nanocomposite	45	*E. coli*, *P. aeruginosa* and *S. aureus*	L929 cell line	The treated group’s organized and developed epithelium was a marked improvement over that of the control group.	[152]
17	AgNPs/Bacterial cellulose composites	10–30 nm	*E. coli*, *S. aureus* and *P. aeruginosa*	Epidermal cells	In vitro studies show that a nanostructured AgNP-BC gel-membrane has the potential to be an effective antimicrobial wound dressing with good biocompatibility for the expedited healing of scald wounds.	[153]
18	Silver NPs embedded bacterial cellulose gel membranes	30	*S. aureus*	Westar rats	A significant amount of healing (85.92%) occurred after 14 days of treatment.	[154]
19	β-chitin-based hydrogels	5	*E. coli* and *S. aureus*	ERO cell line	Manufactured scaffolds showed improved whole-blood clotting ability.	[155]
20	Silver Alginate/Nicotinamide Nanocomposites	20–80	*E. coli* and *S. aureus*	Mice	Significant wound healing had occurred by the fourth day of treatment.	[156]
21	Hyaluronan Nanofiber	25	*E. coli* and *S. aureus*	Cell line (NIH 3T3)	Since nanoparticles are so much smaller than typical particles, they are able to exert a far stronger effect on microbes.	[157]
22	Chitosan-Ag/ZnO composite dressing	10–30 nm	Drug sensitive *E. coli*, *S. aureus* and *P. aeruginosa*	BALB/c mice /L02 cells	These findings support the feasibility of using the prepared chitosan-Ag/ZnO composite dressing in wound care.	[124]
23	Chitosan-based multifunctional hydrogel	250	*E. coli* and *S. aureus*	Rat model	Following 14 days of therapy, the test organism showed the slowest rate of re-epithelialization.	[158]

## Data Availability

The data presented in this study are available on request from the corresponding author.

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
