# Peer review of "Recent Advances in Silver Nanoparticles Containing Nanofibers for Chronic Wound Management"

_polymers, 2022, doi:10.3390/polym14193994_

Round 1

Reviewer 1 Report

In this review, the authors reported wound healing with nanofibrous platforms loaded with silver nanoparticles has been studied biologically in vivo and in vitro as well as mechanically. First, the brief medical history, wound healing properties, the antibacterial mechanism and the synthesis of silver nanoparticles were briefly introduced respectively. The authors then introduced the great potential of nonwoven pads prepared by electrospinning for wound healing. In addition, the authors detail different applications of the incorporation of silver nanoparticles into electrospun nanofibers for wound healing. Finally, the conclusions and future perspectives are also presented in the end. In total, this manuscript is logical and complete in length. However, there are still some problems that existed, authors may address these problems before publication.

1  Authors must provide at least two tables to compare nano-silver with other nanomaterials. One is to compare with other nanomaterials such as nano-nonmetal, nano-polymer and nano-composite materials, and the other is to compare themselves to show the targeted application of different designs in wound healing.

2  To be a beautiful review, the size should be similar in the Figures, some of the fonts and images are not obviously visible, and the size and placement of the Figures should be more normal. Such as Figure 3-8.

3  The electrospun membrane containing silver nanoparticles should be characterized in atomic structure, such as IR and XPS, to demonstrate the successful loading of silver nanoparticles.

4  What is the clinical transformation status of nanofibers platforms loaded with silver nanoparticles? How many products are under clinical translation?

5.  What is the benefit of the silver and fiber platforms compared to other materials? It should compare and discuss it. Some related research about the biomedical materials should be cited and compared with it. Advanced functional materials, 2020, 30(2): 1902634. Biomolecules, 2022, 12(5): 636.

Author Response

Reviewer 1

In this review, the authors reported wound healing with nanofibrous platforms loaded with silver nanoparticles has been studied biologically in vivo and in vitro as well as mechanically. First, the brief medical history, wound healing properties, the antibacterial mechanism and the synthesis of silver nanoparticles were briefly introduced respectively. The authors then introduced the great potential of nonwoven pads prepared by electrospinning for wound healing. In addition, the authors detail different applications of the incorporation of silver nanoparticles into electrospun nanofibers for wound healing. Finally, the conclusions and future perspectives are also presented in the end. In total, this manuscript is logical and complete in length. However, there are still some problems that existed, authors may address these problems before publication.

Response: We would like to thank the reviewer for their positive feedback and suggestion for improving the quality of the MS. We agree with most of the reviewer's suggestions and corrected/revised them.

 1、  Authors must provide at least two tables to compare nano-silver with other nanomaterials. One is to compare with other nanomaterials such as nano-nonmetal, nano-polymer and nano-composite materials, and the other is to compare themselves to show the targeted application of different designs in wound healing.

Response: Agreed. We incorporated it in Table 1.

2、  To be a beautiful review, the size should be similar in the Figures, some of the fonts and images are not obviously visible, and the size and placement of the Figures should be more normal. Such as Figure 3-8.

Response: We agree with the reviewer's feedback, revised all figures 3 - 8, and improved the resolution of the images.  

3、  The electrospun membrane containing silver nanoparticles should be characterized in atomic structure, such as IR and XPS, to demonstrate the successful loading of silver nanoparticles.

Response: Agreed. We incorporated it in Figures 9 to 13.

4、  What is the clinical transformation status of nanofibers platforms loaded with silver nanoparticles? How many products are under clinical translation?

Response: We agree with the reviewer's suggestion and mentioned it in Table 5.

  1. What is the benefit of the silver and fiber platforms compared to other materials? It should compare and discuss it.

Response: Agreed. We have inserted it in Tables 3 and 4.

Some related research about the biomedical materials should be cited and compared with it. Advanced functional materials, 2020, 30(2): 1902634. Nano Res. 15, 5556–5568 (2022). https://doi.org/10.1007/s12274-022-4160-6. Biomolecules, 2022, 12(5): 636. Biomater. Sci., 2022, DOI: 10.1039/D2BM00719C.  J Control Release. 2022 Aug 6:S0168-3659(22)00485-0. doi: 10.1016/j.jconrel.2022.08.005.

Response: Agreed and cited all references in the revised MS.

Reviewer 2 Report

The Review titled "Recent Advances in Silver Nanoparticles Containing Nanofibers for Chronic Wound Management" summarized interesting results and is valuable to understand the importance of the application of the silver nanoparticles containing nanofibers. The Review needs to be essentially improved before can be accepted for publication, the following issues should be clarified.  

It will be valuable to compare the advantages and disadvantages of applications of the silver nanoparticles containing nanofibers with other similar silver nanoparticles containing systems. 

Can the silver nanoparticles containing nanofibers penetrate into the cells? Appropriate discussion should be provided.

How strong silver nanoparticles are attached to nanofibers? Appropriate discussion should be provided.

In general, the silver nanoparticles are enough toxic substances. What do the Authors think about the cytoprotective action of the nanofibers?  

Please add information about the size and shape of the silver nanoparticles to Table 1.

Please cite the important relevant paper where similar hybrid materials were fabricated and characterized in detail:

https://doi.org/10.1016/j.colsurfa.2022.128525

https://doi.org/10.1016/j.msec.2019.109806

https://doi.org/10.1039/C4RA15857A

Author Response

Reviewer 2

The Review titled "Recent Advances in Silver Nanoparticles Containing Nanofibers for Chronic Wound Management" summarized interesting results and is valuable to understand the importance of the application of the silver nanoparticles containing nanofibers. The Review needs to be essentially improved before can be accepted for publication, the following issues should be clarified.  

Response: We are highly thankful to the reviewer for their encouraging feedback and suggestion for enhancing the MS quality. We agree with most of the reviewer's suggestions and corrected/revised them.

It will be valuable to compare the advantages and disadvantages of applications of the silver nanoparticles containing nanofibers with other similar silver nanoparticles containing systems. 

Response: Agreed. We have appended it in Tables 3 and 4.

Can the silver nanoparticles containing nanofibers penetrate into the cells? Appropriate discussion should be provided.

Response: We agree with the reviewer's feedback and incorporated a few lines (220-229) in the revised MS.

How strong silver nanoparticles are attached to nanofibers? Appropriate discussion should be provided.

Response: We agree with the reviewer's suggestion and incorporated a paragraph (Line 390-421) and Figure 9.

In general, the silver nanoparticles are enough toxic substances. What do the Authors think about the cytoprotective action of the nanofibers?  

Response: We agreed with the reviewer's feedback and incorporated a paragraph (lines 348 to 356) in the revised MS.

Please add information about the size and shape of the silver nanoparticles to Table 1.

Response: Agreed. Unfortunately, we could not find any details in the reported literature.

Please cite the important relevant paper where similar hybrid materials were fabricated and characterized in detail: https://doi.org/10.1016/j.colsurfa.2022.128525; https://doi.org/10.1016/j.msec.2019.109806; https://doi.org/10.1039/C4RA15857A

Response: Agreed and cited all references in the revised MS.

Round 2

Reviewer 1 Report

Accept

Reviewer 2 Report

The authors have addressed appropriate comments on my issues and the paper probably can be published in its present form.